# Oral Cancer and Sleep Disturbances: A Narrative Review on Exploring the Bidirectional Relationship

**DOI:** 10.3390/cancers17081262

**Published:** 2025-04-08

**Authors:** Runhua Yang, Hongyu Jin, Chenyu Zhao, Wei Wang, Wen-Yang Li

**Affiliations:** 1Respiratory and Critical Care Department, The First Hospital of China Medical University, Shenyang 110001, China; yrh4723889@163.com (R.Y.); wwbycmu@126.com (W.W.); 2Department of China Medical University-The Queen’s University of Belfast Joint College, School of Pharmacy, China Medical University, Shenyang 110052, China; czhao04@qub.ac.uk

**Keywords:** oral cancer, sleep disturbances, bidirectional relationship, obstructive sleep apnea, intermittent hypoxia, disruption of circadian rhythms

## Abstract

There may be a complex bidirectional relationship between oral cancer and specific sleep disorders, such as insomnia and obstructive sleep apnea (OSA). The symptoms and treatment of oral cancer can affect the patient’s sleep status, while sleep disorders may also exacerbate tumor progression through mechanisms such as chronic inflammation and hypoxia. A deeper exploration of this relationship could offer novel strategies for the multidisciplinary management of oral cancer patients, integrating sleep health into therapeutic protocols.

## 1. Introduction

Oral cancer is a malignant tumor derived from the squamous epithelial cells of the oral mucosa, also known as oral squamous cell carcinoma (OSCC) [1]. According to GLOBOCAN 2020, the incidence of oral cancer accounts for 2.6% of all cancers worldwide, with approximately 400,000 people diagnosed with oropharyngeal cancer each year [2]. Due to the hidden onset of oral cancer, coupled with its high incidence and mortality rates, patients often face multiple psychological and physiological challenges. Among these, changes in sleep structure and the increased prevalence of sleep disorders are primary factors affecting the quality of life for patients [3]. Studies have shown that 44% of 560 newly diagnosed patients with oral cancer experience poor sleep quality, which is significantly higher than the incidence of sleep disorders in the general population [4]. Moreover, even among patients who have survived for over a year after being cured of oral cancer, a substantial proportion still experiences varying degrees of sleep disorders, with respiratory-related sleep disorders, specifically obstructive sleep apnea (OSA), being particularly prevalent [3]. Among oral cancer patients, especially those who have undergone radiotherapy and surgical treatment, the incidence of OSA is significantly higher. Many oral cancer patients may have underlying sleep apnea even if they do not exhibit typical OSA symptoms in clinical practice (such as snoring, daytime sleepiness, etc.) [5]. Another study demonstrated that among 23 patients newly diagnosed with oral cancer who underwent free flap reconstruction surgery, the likelihood of developing OSA increased from 91.3% to 95.6%, with the apnea-hypopnea index (AHI) significantly increasing at 6 months after surgery [6].

Sleep is an important factor affecting human health, and the normal sleep structure is primarily divided into non-rapid eye movement sleep and rapid eye movement sleep (REM). The series of abnormal phenomena that occur during sleep are called sleep disorders, with common types including insomnia, obstructive sleep apnea (OSA), circadian rhythm sleep disorders, episodic sleep disorder, and cataplexy syndrome. Other abnormal sleep disorders (such as partial arousal disorder and REM behavior disorder) have been shown to be associated with adaptive immune and inflammatory regulation and are closely linked to the occurrence and recurrence of tumors [7]. Research has demonstrated that compared to individuals who sleep 7–8 h per day, those who sleep less than 6 h face a greater cancer risk, with an increase of over 40% [8]. Additionally, OSA can affect immune system function, increase oral mucosal fragility, exacerbate oral mucosal inflammation in patients with oral cancer, and lead to a decline in quality of life [9]. One study analyzed saliva samples from 30 children with obstructive sleep apnea (OSA) and 30 normal children using high-throughput sequencing. It was found that the oral microbiota diversity of OSA children was lower, and the microbiota related to inflammation and immunity was significantly enriched. This suggests that OSA may affect oral immunity and inflammation processes by altering the oral microbiota [10]. The primary pathophysiological mechanisms of OSA include chronic inflammation and oxidative stress, both of which are critical factors influencing the development and progression of oral cancer. Chronic inflammation is strongly associated with the initiation, metastasis, and invasion of oral cancer [11], likely through interactions among various pro-inflammatory mediators. For instance, epidermal growth factor (EGF) can induce epithelial–mesenchymal transition (EMT) in oral cancer cells via the Akt/Ezrin Tyr353/NF-κB pathway, thereby enhancing their motility and invasiveness [12]. Multiple studies have identified matrix metalloproteinases (MMPs) as key mediators of oral cancer progression. Specifically, MMP-11 has been linked to lymph node metastasis and increased malignancy, with high MMP-11 expression correlating with significantly reduced survival rates in oral cancer patients [13]. The inflammatory cytokine interleukin-11 (IL-11)/gp130 can upregulate MMP-13 expression in oral squamous cell carcinoma (OSCC) through the activation of the PI3K/Akt and AP-1 signaling pathways, thereby promoting cancer cell migration [14]. Furthermore, oxidative stress contributes to oral cancer development by promoting M1-like tumor-associated macrophage polarization via Thbs1-mediated mechanisms [15].

So far, multiple lines of evidence have indicated a bidirectional relationship between sleep disorders, particularly OSA, and oral cancer. This study is a narrative review that primarily integrates research evidence from the PubMed database on oral cancer and sleep disturbances from 1994 to 2025. The search strategy employed the following keyword combinations: “sleep disturbances”, “obstructive sleep apnea syndrome”, “oral cancer”, “oral squamous cell carcinoma”, “etiology of oral cancer”, “etiology of sleep disturbances”, “treatment of oral cancer”, and “treatment of sleep disturbances”. The literature selection primarily focuses on studies published in the last decade, with the research topic required to involve oral cancer (especially oral squamous cell carcinoma, OSCC) or sleep disturbances (such as obstructive sleep apnea, OSA), including their etiology, pathogenesis, treatment, and correlations. Additionally, early seminal studies of significant importance are selectively included. Studies that are unrelated to the research topic or of low methodological quality such as those with small sample sizes, flawed study designs, or unsupported conclusions are excluded. This article focuses on the clinical manifestations, pathological mechanisms, and potential clinical interventions and treatments related to sleep disorders and oral cancer. It analyzes the multifaceted interactions between oral cancer and sleep, discusses aspects that have been overlooked or inadequately addressed in existing research, and explores new directions for the treatment and management of oral cancer.

## 2. Mechanisms by Which Oral Cancer Leads to Sleep Disturbances

### 2.1. Tumor Size and Location

Sleep disorders are commonly observed in patients with oral cancer. Studies have shown that among 412 patients diagnosed with oral cancer, approximately one-third experience persistently poor or worsening sleep quality [16]. Moreover, the prevalence of obstructive sleep apnea (OSA) in this population is as high as 91.7%, significantly higher than in the general population. This phenomenon may be associated with the tumor’s size and anatomical location [17]. Common symptoms of oral cancer include persistent ulcers, lumps, oral pain, dysphagia, and tongue or gingival bleeding. As the tumor develops, cervical lymph nodes may swell and invade adjacent tissues and organs. Oral cancer can occur in any part of the mouth, including the tongue, the floor of the mouth, gingiva, cheek, hard palate, soft palate, and more [18]. In general, patients whose tumors are located in the upper airway (UA), such as the oropharynx, laryngopharynx, or nasal cavity, are more likely to exhibit OSA symptoms [19]. This may be because when the tumor causes swelling, enlargement, and backward displacement of the tongue, it partially blocks the larynx or UA. Additionally, when the tumor invades the soft palate and oropharynx, it causes airway stenosis. The enlarged lymph nodes may also compress UA muscles and tissues, damaging their function and leading to OSA. Studies have shown that the size of tumors in patients with oral cancer is significantly correlated with the severity of OSA. Specifically, the larger the tumor volume, the higher the risk of OSA in patients. This is generally due to the direct compression of the tumor on the UA, and larger tumors lead to more severe pharyngeal obstruction [20].

### 2.2. Cancerous Pain and Discomfort

Patients with oral cancer usually experience severe pain, with both the incidence and intensity of pain being higher than those observed in other cancers [21]. Cancer pain refers to pain caused either by the cancer itself or its treatment (such as surgery, radiotherapy, chemotherapy, etc.). The pain caused by the tumor itself can generally be divided into two categories: physical pain caused by the compression and erosion of surrounding tissues and nerve injury pain resulting from the interaction between tumor cells and immune cells. Pain often varies with tumor size, location, and individual differences [22]. For oral cancer patients, orofacial pain is one of the earliest and most severe symptoms, impairing their speech, swallowing, eating, and drinking functions [21]. A recent study showed that tumor necrosis factor-alpha (TNF-α), secreted by oral cancer cells, can affect other cell types in the tumor microenvironment, such as increasing the infiltration of immune cells (like T cells), sensitizing trigeminal lingual afferent neurons to enhance the transmission of pain signals, and driving the release of inflammatory mediators, which increases the intensity and duration of cancer pain [23]. Protease-activated receptor 2 (PAR2) on sensory neurons is a key factor in pain signaling. Studies have found that growing oral cancer tumor cells can release proteases, such as cathepsin S, to activate PAR2, thereby stimulating sensory neurons in surrounding tissues [24]. In addition, oral cancer cells can also release brain-derived neurotrophic factor (BDNF), which activates TrkB receptors in peripheral nerves, leading to nerve hypersensitivity and exacerbating pain in patients with oral cancer [25].

Studies have shown that severe and persistent chronic pain often leads to a decline in sleep quality [26]. Moreover, patients suffering from chronic facial pain tend to experience reduced sleep duration, lower health-related quality of life, and increased levels of stress and fatigue [27].

In addition to cancer itself, cancer treatments (surgery, radiotherapy, and chemotherapy) can also cause acute and chronic pain in patients [28]. Chronic pain is often more easily overlooked and difficult to alleviate with conventional analgesic drugs. It is generally considered a long-term interference factor affecting patients’ quality of sleep and significantly reduces their treatment compliance [29]. A recent cross-sectional study reported that radiotherapy in patients with oral cancer frequently leads to chronic neuropathic pain, with a prevalence of 67.1%. The pain is mainly manifested as tingling, burning, and numbness in the oral and laryngeal regions, significantly increasing the risk of sleep disturbances, such as insomnia [30]. In the study of quantitative assessment of progressive pain in patients with oral cancer who received radiotherapy, it was also found that, compared to healthy subjects, radiotherapy patients not only suffered from chronic pain but also exhibited reduced pain pressure thresholds, upper limb dysfunction, and extensive hyperalgesia, suggesting that cancer treatment may lead to the overactivation of the nervous system, a phenomenon known as central sensitization [31]. Another longitudinal qualitative study on pain caused by radiotherapy in patients with oral cancer indicated that this pain is long-term and severely impacts the patients’ daily lives, particularly their sleep [32]. For patients with oral cancer, cancer pain is a significant factor contributing to sleep disorders, such as insomnia. Studies have shown that patients with acute and chronic pain, chronic orofacial pain, and chronic physical pain generally experience poor sleep quality, with pain intensity being positively correlated with poor sleep quality [33]. Some studies have analyzed the factors leading to poor sleep quality in newly diagnosed oral cancer patients, highlighting that oral pain is significantly associated with poor sleep [4]. Polysomnographic analysis of patients with acute and chronic pain generally shows reduced slow-wave and REM sleep periods, suggesting that pain causes nighttime awakenings, sleep shortening, and fragmentation [34]. The underlying mechanism of pain-induced sleep disturbances, such as insomnia, may involve increased activity in locus coeruleus norepinephrine neurons, which release norepinephrine, triggering an arousal response, disturbing the normal sleep cycle, and ultimately affecting sleep structure and quality [35]. Additionally, chronic pain may cause systemic chronic low-grade inflammation, which is closely related to immune responses [36], and many cytokines have been shown to regulate physiological sleep in animals [37].

### 2.3. Psychological Factors

Patients with oral cancer often experience varying degrees of mental health problems and emotional distress [38]. During the diagnosis and treatment of oral cancer, anxiety and depression can be triggered by worries about physical changes, treatment-related pain, and fear of the future. Both are common mental burdens for oral cancer patients, which not only affect their physical health but also hinder their treatment compliance [39], According to surveys, even survivors of oral cancer exhibit higher levels of anxiety and depression compared to the general population [3]. Another study assessing the psychological and social functions of oral cancer patients post-chemotherapy revealed that over 35% still experience long-term sequelae, including fatigue and cognitive decline, which contribute significantly to anxiety and depression [40]. Psychological factors frequently contribute to sleep disorders in these patients. One study demonstrated that the HADS anxiety and HADS depression scores of oral cancer patients were significantly correlated with sleep quality, sleep latency, sleep disorders, and daytime dysfunction. Notably, 82% of oral cancer patients did not receive psychological support therapy [41]. Recent research has shown that these psychoneurological symptoms are positively correlated with inflammatory markers, such as IL-6 and C-reactive protein, in oral cancer patients, and are strongly linked to poor sleep quality and fatigue [42]. Additionally, another study indicated that changes in the expression of neurotransmitter-related genes, such as BDNF and COMT, in oral cancer patients may be associated with sleep disorders and anxiety [43].

### 2.4. Treatment-Related Side Effects

In addition to directly causing OSA through tumor compression and the invasion of the upper airway, all treatment methods for oral cancer, including surgery, radiotherapy, and chemotherapy, can also induce OSA [41]. The impact of surgery on the risk of OSA in oral cancer patients is closely linked to the surgical technique, specifically the choice of microvascular flap and tumor location. Radical surgery for primary oral squamous cell carcinoma typically involves the removal of the tumor and its surrounding tissues, followed by free flap reconstruction. This process may alter the tissue’s flexibility, potentially leading to stenosis in the oral cavity, pharynx, and airway [44]. Notably, removing tumors from the back of the mouth, soft palate, or pharynx can directly compromise the airway’s integrity, resulting in severe posterior airway stenosis [45]. Studies indicate that oral cancer patients undergoing radial forearm free flap reconstruction are at a high risk of developing moderate to severe OSA [46]. This is because, while the forearm flap excels in oral cavity reconstruction, it plays a limited role in restoring pharyngeal function and supporting the posterior airway. In contrast, larger muscle flaps, such as the latissimus dorsi flap, offer enhanced muscle support during pharyngeal and soft palate area reconstruction, thereby minimizing posterior airway alterations and reducing the likelihood of airway stenosis. Consequently, this approach helps lower the risk of OSA [45].

In the process of radiotherapy for oral cancer, radiation often involves the pharynx, neck, and throat [47]. Studies have found that radiotherapy can result in changes to the anatomical structure of these areas, which increases the probability of airway collapse and stenosis in patients. Additionally, radiotherapy reduces the tension and elasticity of the upper airway (UA) muscles, impairs UA stability, and subsequently causes UA obstruction, apnea, snoring, and sleep disorders [48]. Therefore, the risk of OSA in patients with oral cancer following radiotherapy is significantly higher [49]. Among oral cancer patients, even though the body mass index (BMI) is significantly reduced and the retrolingual pharyngeal area increases after radiotherapy, the prevalence of OSA remains higher than in others [50]. Moreover, side effects of radiotherapy for oral cancer, such as oral ulcers and pain, frequently lead to sleep disturbances, such as insomnia, and a decline in quality of life for patients [51].

Chemotherapy is commonly used in the adjuvant treatment of patients with oral cancer. Some studies evaluated the sleep patterns of patients with oral cancer undergoing palliative chemotherapy through the Fitbit Charge 4 sleep tracking device. It was found that during chemotherapy, patients generally experienced a decline in sleep quality, such as nighttime awakening and shortened total sleep time [52]. The key mechanism may be that chemotherapy causes circadian rhythm disruption by affecting the expression of core clock genes (such as *BMAL1*, *Clock*, *PER*, and *CRY*) and the levels of hormones (such as cortisol and melatonin) [53], which leads to daytime sleepiness, fatigue, and an increased risk of OSA in patients with oral cancer.

## 3. Potential Mechanisms Underlying the Correlation Between Sleep Disorders and Oral Cancer Occurrence

### 3.1. Intermittent Hypoxia and Tumor Occurrence

Multiple studies have shown a significant correlation between sleep disorders, especially OSA, and the occurrence of oral cancer. Patients diagnosed with insomnia and OSA have a higher probability of developing oral cancer [54], OSA is mainly manifested as recurrent upper airway obstruction and hypoxemia, characterized by intermittent hypoxia and fragmented sleep [55]. It is generally believed that intermittent hypoxia is the main factor leading to cancer. Recent studies have shown that intermittent hypoxia can activate the hypoxia-inducible factor (HIF-1) signaling pathway, leading to the increased expression of downstream genes (such as *VEGF*, *MMPs*, *EPO*, etc.), promoting tumor angiogenesis, cell proliferation, and metabolic reprogramming, providing conditions for the growth and invasion of cancer cells. It can also upregulate the expression of pro-cancer miRNAs (such as *miR-210*), improve the ability and resistance of cancer cells to proliferation and invasion, downregulate the expression of tumor suppressive miRNAs, and further promote intermittent hypoxia’s promoting effect on tumors [56]. From the perspective of inflammation and immune suppression, intermittent hypoxia activates multiple inflammatory pathways, including nuclear factor kappa B (NF-κB) and pro-inflammatory cytokine (IL-6, IL-8) signaling pathways, causing chronic inflammation throughout the body and providing a microenvironment for tumor development. Additionally, NF-κB is a critical regulator of metastasis in oral cancer, primarily mediating the epithelial–mesenchymal transition (EMT) in cancer cells. Studies have shown that *ZEB1/2*, Transforming Growth Factor-beta (TGF-β), and Slug act as EMT inducers upregulated by NF-κB. Following NF-κB-driven EMT induction, E-cadherin levels are significantly reduced, while N-cadherin and vimentin levels are markedly elevated, thereby promoting the metastasis of tumor cells [57].

In addition, intermittent hypoxia can lead to oxidative stress imbalance, increase the production of lipid peroxides (such as malondialdehyde (MDA)), and reduce the activity of antioxidant enzymes (such as superoxide dismutase (SOD)), thereby promoting DNA and tissue damage and increasing the risk of tumor development [58].

Studies have shown that hypoxia also has a significant impact on the effect of radiotherapy and chemotherapy for oral cancer. Tumor cells in hypoxic areas are usually insensitive to radiotherapy and chemotherapy. Tumor cells in a hypoxic environment can improve their tolerance to radiotherapy and chemotherapy by changing cell cycle progression and increasing antioxidant capacity [59]. Therefore, hypoxia seriously affects the survival outcome of patients with oral cancer, and severe nocturnal hypoxemia almost triples the mortality of all cancers [60]

### 3.2. Dysregulation of Circadian Rhythm and Tumor Progression

Circadian rhythm refers to the 24 h periodic changes regulated by the biological clock, which is controlled by the suprachiasmatic nucleus (SCN) in the brain. It can adjust the biological clock according to external light and environmental changes and has a significant impact on physiological processes such as hormone secretion, metabolism, and immune function. Most drug effects are also affected by circadian rhythm. Dysregulation of the circadian rhythm is thought to promote tumorigenesis by promoting cell proliferation, inhibiting apoptosis, and enhancing the viability of cancer cells [61]. A recent study found that the expression of multiple circadian genes was significantly altered in the tumor tissues of patients with oral squamous cell carcinoma. For example, the expression of core genes responsible for regulating the circadian clock (such as *Clock*, *Bmal1*, *Per*, and *Cry*) tends to decline in oral cancer, while some tumor-related genes (such as *Reverb*) are upregulated, indicating that circadian rhythm is dysregulated. In particular, changes in the *PER3* gene may directly accelerate tumor development by promoting the excessive proliferation of tumor cells and enhancing their anti-apoptotic ability [62]. On the other hand, circadian rhythm has recently been found to affect the proliferation and differentiation of cancer stem cells (CSCs) by regulating the expression of key genes (such as cyclin activators). The specific mechanism is as follows: the core clock gene Period 1 (*PER1*) can inhibit glycolysis-mediated cell proliferation by forming the PER1/RACK1/PI3K complex, thereby suppressing the progression of oral cancer cells [63]. Another study also suggests that the clock gene Per2 plays a crucial role in the balance of cell cycle progression, proliferation, and apoptosis by regulating the cyclin/CDK/CKI cell cycle network [64]. Cancer stem cells are key factors in tumor self-renewal, metastasis, and drug resistance. Circadian rhythm also affects the tumor microenvironment, influencing cell metabolism, immune response, angiogenesis, and oxygenation levels to promote tumor growth and metastasis [65].

## 4. Other Shared Underlying Mechanisms in This Bidirectional Relationship

### 4.1. Endocrine Immune Factors

The neuroendocrine system is closely related to the occurrence, development, and metastasis of tumors through neurotransmitters, hormones, and signaling molecules [66]. Catecholamines are chemical substances secreted by the adrenal gland and nervous system, including epinephrine, norepinephrine, and dopamine. It has been confirmed that catecholamines are primarily involved in regulating the stress response and act as stress hormones that induce tumorigenesis. The cancer incidence in patients using β-blockers is significantly reduced [67]. A study has shown that in oral cancer patients, elevated levels of adrenaline and norepinephrine are significantly associated with increased anxiety scores and poor sleep quality. Elevated catecholamine levels may affect tumor progression by promoting inflammation and angiogenesis in the tumor microenvironment, suggesting that sleep disturbances, such as insomnia, and anxiety in oral cancer patients may influence disease progression and prognosis through neuroendocrine pathways [68]. Another study suggests that activation of the sympathetic nervous system caused by stress in oral cancer patients may impact the tumor microenvironment through the β1- and β2-AR signaling pathways, thereby promoting tumor cell invasion and metastasis [69].

In addition, recent studies have shown that salivary melatonin levels in oral cancer patients are generally lower than those in healthy controls [70]. Melatonin, as a regulating hormone for sleep and wakefulness, has anti-inflammatory, antioxidant, and immunomodulatory effects, which may be related to tumor progression, immune suppression, and treatment-related side effects [71]. Its protective mechanism for oral health can be categorized into regulating biological rhythms, oxidative stress responses, inflammatory responses, and other physiological processes through MT1 and MT2 receptors, thereby regulating oral immune responses. Additionally, melatonin participates in regulating oral tissue regeneration and repair through MT3 and RZR/ROR receptors, playing a potentially positive role in the proliferation, differentiation, and tissue repair of oral cells [72]. Moreover, melatonin can inhibit cancer cell proliferation by preventing G1/S and G2/M transitions and interacting with cyclins and their dependent kinases (CDKs). Under certain conditions, melatonin can interact with autophagy-related proteins (such as LC3, Beclin-1, mTOR), induce autophagy, remove harmful cellular components, and protect normal cells. By increasing the activity of pro-apoptotic proteins (such as Bax) and reducing the expression of anti-apoptotic proteins (such as Bcl-2), melatonin can promote the initiation of the endogenous apoptotic pathway and induce cancer cell apoptosis in vivo [73].

The effects of melatonin in inhibiting cancer cell proliferation, promoting autophagy, and inducing apoptosis not only contribute to the suppression of oral cancer progression but may also influence the overall physiological state of the body, particularly sleep regulation. As a key hormone regulating the sleep–wake cycle, abnormal melatonin levels are often closely associated with sleep disorders. Studies have shown that exogenous melatonin supplementation can be used to treat primary and secondary sleep disorders, reduce sleep latency in primary insomnia, and improve delayed sleep phase syndrome, thereby optimizing the sleep–wake pattern [74]. Given that oral cancer patients often experience reduced sleep quality, circadian rhythm disturbances, and impaired immune function, appropriate melatonin supplementation may not only improve their sleep conditions but also exert anti-inflammatory, immunomodulatory, and antitumor effects. Cellular experiments have demonstrated that melatonin, in combination with the ferroptosis inducer erastin, can exert a synergistic antitumor effect on oral squamous cell carcinoma by inducing apoptosis, ferroptosis, and inhibiting autophagy through ROS promotion [75]. Additionally, research has found that exogenous supplementation of melatonin and vitamin D can help address antioxidant imbalance in oral cancer. Melatonin supplementation may inhibit tumor growth, improve survival rates and quality of life, and enhance the efficacy of radiotherapy [76], thereby providing an adjunctive strategy for oral cancer treatment.

### 4.2. Lifestyle

Smoking is considered a major risk factor for both oral cancer and obstructive sleep apnea (OSA). Carcinogens in tobacco can directly affect cells in the mouth, increasing the likelihood of carcinogenesis [77]. Even mild exposure to secondhand smoke has been shown to elevate the risk of oral cancer [78]. The molecular mechanism behind this involves carcinogens in smoking causing DNA damage and gene mutations through oxidative stress and free radical generation, particularly mutations in tumor suppressor genes like *p53*, which promote tumorigenesis. Additionally, smoking may enhance cancer cell proliferation by activating oncogenes (such as *k-ras*) or inhibiting tumor suppressor genes (such as *p16INK4a*). Smoking can also alter gene expression via epigenetic mechanisms, such as DNA methylation and histone modification, leading to the onset of oral cancer [79]. Furthermore, smoking is strongly associated with the development of OSA. It aggravates upper airway (UA) obstruction by increasing airway inflammation and edema. Smokers’ airways are more prone to collapse than those of non-smokers, which can trigger sleep apnea [80].

Nutrition plays a crucial role in both oral cancer and obstructive sleep apnea (OSA). Patients with oral cancer often face the risk of inadequate nutrient intake due to the presence of the tumor itself and the side effects that may arise during treatment, such as oral ulcers, dry mouth, taste alterations, nausea, and loss of appetite caused by surgery, radiotherapy, or chemotherapy. Additionally, difficulties in chewing and swallowing further limit their ability to consume essential nutrients, making them more susceptible to malnutrition. Studies have shown that nutritional and physical prehabilitation interventions can improve the overall health status of oral cancer patients and positively impact clinical outcomes [81]. Malnutrition not only leads to weight loss, impaired immune function, and muscle wasting but may also negatively affect sleep quality. For instance, vitamin D deficiency has been associated with an increased risk of OSA. The underlying mechanism may involve vitamin D receptors and the enzymes that regulate its activation and degradation, which are implicated in sleep regulation in the brain. Furthermore, vitamin D plays a role in the pathway for melatonin production, a hormone that regulates the human circadian rhythm and sleep. In addition, vitamin D may indirectly influence sleep through its effects on nonspecific pain disorders and has been linked to obstructive sleep apnea syndrome [82].

Recent studies have indicated that obesity is closely linked to an increased risk of developing oral cancer. For individuals with a high body mass index (BMI), obesity can facilitate the onset and progression of oral cancer through complex biological mechanisms. On one hand, obesity causes hormone imbalances, including alterations in insulin, fatty acids, and lipids, particularly an increase in insulin-like growth factor (IGF), which promotes tumor cell proliferation. Obesity also triggers chronic low-grade inflammation throughout the body, further enhancing tumor cell proliferation and promoting metastasis. On the other hand, the adipose tissue in obese individuals releases cytokines and fatty acids that alter the tumor microenvironment, fostering tumor growth and spread, diminishing immune cell activity, and increasing tumor cell resistance to treatments. This leads to reduced efficacy in radiotherapy, chemotherapy, and immunotherapy, and heightens the risks and side effects of these treatments [83]. The adipokine Chemerin can enhance the invasiveness of oral cancer cells by activating the STAT3 signaling pathway and promoting the production of IL-6 and TNF-α [84]. Additionally, high-fat diet (HFD) induced obesity significantly promotes the occurrence of oral cancer. The underlying mechanism may involve obesity recruiting CD11bGr1 myeloid-derived suppressor cells (MDSCs) through the CCL9/CCR1 axis, thereby altering the local immune microenvironment and enhancing the immunosuppressive function of MDSCs via intracellular fatty acid uptake [85]. Recent research has also revealed that RNA methylation regulated by the fat mass and obesity-associated protein (FTO) gene plays a crucial role in tumor cell proliferation, migration, and invasion. The FTO gene, which is involved in fat metabolism, may contribute to the development and progression of oral cancer by influencing tumor cell metabolism and inhibiting the expression of tumor suppressor genes [86].

Over 50% of OSA patients are obese [87]. A Mendelian study has shown a causal relationship between BMI, body fat mass, and the occurrence of OSA [88]. The underlying cause of obesity-induced OSA is the increased fat accumulation around the throat and pharynx in obese individuals, which compresses the UA, making it more prone to collapse compared to healthy individuals. Studies have shown that obese individuals exhibit heightened sympathetic nerve activity and diminished parasympathetic nerve function, resulting in reduced airway tension and further exacerbating the collapse of the UA. Additionally, an increase in body weight elevates lung load, alters respiratory patterns, and indirectly contributes to the development of OSA [89]. Cytokines secreted by adipose tissue, known as adipokines, primarily include leptin, adiponectin, and inflammatory cytokines. Research indicates that adipokines can not only disrupt normal metabolism through inflammatory responses but also regulate the central nervous system and sleep cycle. Obese individuals tend to have abnormal adipokine secretion, which indirectly contributes to the onset and progression of OSA [89].

## 5. Intervention and Treatment Measures

### 5.1. Lifestyle Management and Psychobehavioral Therapy

A multidisciplinary approach to treating sleep disorders in oral cancer patients is essential (as illustrated in the Figure 1). A study using the National Health and Nutrition Examination Survey (NHANES) database to investigate the correlation between dietary intake and sleep disturbances, such as insomnia, in cancer survivors found that adequate dietary fiber intake may be an effective strategy to alleviate sleep problems in oral cancer survivors [90]. This suggests that adopting a healthy eating pattern can significantly contribute to improving sleep quality. A balanced diet should focus on avoiding high-sugar and high-fat foods while increasing the intake of foods rich in antioxidants, vitamins, and minerals, such as fruits, vegetables, seeds, and lean proteins. Additionally, establishing good eating habits, such as avoiding large meals at night, is crucial for overall health. A meta-analysis of case studies, single-group designs, non-randomized controlled trials, and one randomized controlled trial highlighted the benefits of psychological interventions, such as relaxation training and meditation, in improving the quality of life for oral cancer patients. These interventions have been shown to be particularly effective in alleviating sleep disturbances, such as insomnia, and anxiety [91]. Furthermore, a randomized controlled trial on the impact of behavioral therapy on postoperative pain, fatigue, and vital signs in oral cancer patients found that progressive muscle relaxation (PMR) significantly reduced sleep disturbances, such as insomnia, pain, fatigue, muscle tension, anxiety, and depression in patients undergoing major surgery [92]. Studies have shown that a 12-week exercise intervention, including strength training, resistance training, walking, cycling, yoga, qigong, or tai chi, can significantly improve sleep disorders, anxiety, pain, and fatigue in patients with oral cancer [93]. Another study also indicated that regular physical activity interventions help improve cardiopulmonary health, fatigue levels, and overall quality of life in oral cancer survivors [94]. Enhancing physical activity may be a key approach to improving sleep quality in oral cancer patients.

### 5.2. Pain Management and OSA Treatment

In addition to surgery and radiation/chemotherapy, comprehensive treatment for oral cancer patients must also include pain management. Common analgesic medications for pain relief include non-steroidal anti-inflammatory drugs (NSAIDs), opioids, local anesthetics, and neuromodulators. For sleep disturbances, such as insomnia, stemming from psychological factors like anxiety or depression, antidepressants and anxiolytics are primarily used [95]. A double-blind study comparing piroxicam and acetylsalicylic acid for pain management in oral cancer patients found that both drugs effectively reduced pain and improved sleep duration [96]. Regarding obstructive sleep apnea (OSA), continuous positive airway pressure (CPAP) is the clinical treatment of choice for oral cancer patients. A clinical study examining the use of CPAP for OSA in oral cancer patients after concurrent chemoradiation found that CPAP had positive effects not only on cardiovascular and metabolic systems but also on survival rates for oral cancer patients [97]. Due to subjective discomfort, including dry mouth and chest discomfort, CPAP adherence is poor. Even among patients who initiate CPAP therapy, 50% discontinue its use within the first year [98]. Therefore, finding alternative treatments for CPAP is crucial. Currently, orofacial myofunctional therapy (OMT), which includes isotonic and isometric exercises targeting the oral and oropharyngeal structures, has been clinically adopted. OMT can improve clinical symptoms in OSA patients by enhancing muscle tone, endurance, and coordination of the pharyngeal and peripharyngeal muscles. Benefits include reducing snoring, lowering the apnea-hypopnea index, and improving daytime sleepiness and sleep quality [99]. Additionally, positional therapy, which involves devices that prevent the supine position during sleep by inducing neck or chest vibrations, has shown significant efficacy in OSA patients who experience more frequent and severe respiratory events while sleeping in the supine position (postural related sleep apnea) [100]. For certain patients, oral appliances and airway implants, such as tongue muscle stimulators, may offer effective alternative treatment options. However, there is currently no specific medication available that effectively treats OSA in these patients [100].

## 6. Conclusions and Future Research Directions

Based on the findings of this research, it is evident that there exists a complex, bidirectional relationship between sleep disorders, particularly OSA, and oral cancer. Psychological factors, including severe pain, anxiety, depression, and the side effects of radiotherapy and chemotherapy, contribute to the onset of sleep disorders in oral cancer patients through inflammatory mechanisms, gene expression changes, and alterations in hormone levels. Conversely, sleep disorders, primarily OSA, exacerbate the psychological burden on patients, impacting mood and emotional wellbeing. Intermittent hypoxia, circadian rhythm disruption, and hormonal imbalances associated with sleep disturbances can interfere with treatment progress, promote tumor growth, and hinder recovery. This interplay not only affects the physical health of patients but also their psychological state and quality of life, creating a vicious cycle that complicates the overall management of the disease (as illustrated in the Figure 2).

However, this narrative review has certain limitations. First, the study relies on previously published literature, which may introduce selection and publication bias. The heterogeneous nature of the included studies, including differences in study design, sample population, and sleep assessment tools, poses a challenge for direct comparisons. Additionally, most studies are retrospective or observational, limiting causal inference. Biases in the included studies, such as recall bias, confounding factors, and small sample sizes, may also affect the generalizability of the findings. Future research should focus on large-scale prospective cohort studies with objective sleep assessment and better control for confounders to establish a clearer causal relationship between oral cancer and sleep disturbances.

Current evidence suggests a potential association between obstructive sleep apnea (OSA) and oral cancer; however, a direct causal relationship remains unclear. This association may be mediated through various mechanisms, including chronic intermittent hypoxia, oxidative stress, and inflammatory responses. However, the presence of shared risk factors, such as smoking, obesity, and chronic inflammation, complicates the determination of causality. Regarding longitudinal studies, while some prospective cohort studies have indicated an increased incidence of certain cancers among OSA patients, specific data on oral cancer remain limited. Long-term follow-up studies are needed to establish a causal relationship. Future research should focus on large-scale longitudinal cohorts and interventional trials to explore whether OSA treatments, such as continuous positive airway pressure (CPAP), can reduce cancer risk.

Despite the growing evidence of this relationship, current research largely focuses on the surface-level associations between sleep disturbances and oral cancer. There is a notable gap in understanding the biological mechanisms through which sleep disorders directly influence immune escape, the tumor microenvironment, and tumor progression in oral cancer patients. Future research should focus on the following key areas: at the mechanistic level, further investigations are needed to elucidate the regulatory pathways through which circadian rhythm disruption, inflammatory responses, immune dysregulation, and hormonal fluctuations influence tumor growth and metastasis. Large-scale, multicenter prospective cohort studies should be designed to establish the causal and temporal relationships between sleep disturbances and the incidence or prognosis of oral cancer. Additionally, interventional clinical trials are essential to assess the impact of sleep optimization strategies—such as continuous positive airway pressure (CPAP) therapy and cognitive behavioral therapy—on tumor progression. Epigenetic mechanisms warrant particular attention, with a focus on exploring sleep disturbance-related alterations in DNA methylation and histone modifications as potential biomarkers. From a translational perspective, developing risk prediction models that integrate genetic profiles, metabolic phenotypes, and lifestyle factors will be crucial for advancing personalized sleep management. Moreover, a comprehensive evaluation of sleep interventions on patients’ psychological wellbeing and quality of life is necessary to establish a multidimensional bio-psycho-social support system.

Further investigation is needed to explore how optimizing sleep quality and alleviating OSA symptoms may improve the treatment outcomes for oral cancer patients. Additionally, research should focus on the impact of individualized treatment strategies, considering sleep conditions, on the overall therapeutic approach.

For example, as OSA tends to worsen during radiotherapy and chemotherapy for oral cancer, it becomes crucial to determine the next steps for comprehensive management during these treatments.

Therefore, optimizing sleep patterns and addressing OSA should become integral components of oral cancer treatment plans. A single treatment modality is insufficient to meet the multifaceted challenges posed by oral cancer. A more holistic, interdisciplinary approach is necessary, combining tumor treatment with psychological support, cognitive-behavioral therapy, and sleep optimization. This integrated strategy will help create a personalized, comprehensive treatment plan, leading to improved treatment efficacy and quality of life for oral cancer patients.

## Figures and Tables

**Figure 1 cancers-17-01262-f001:**
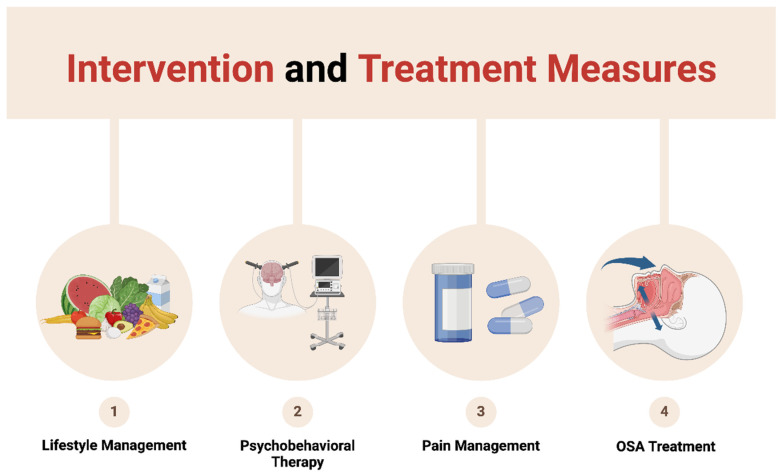
Interventions and treatments.

**Figure 2 cancers-17-01262-f002:**
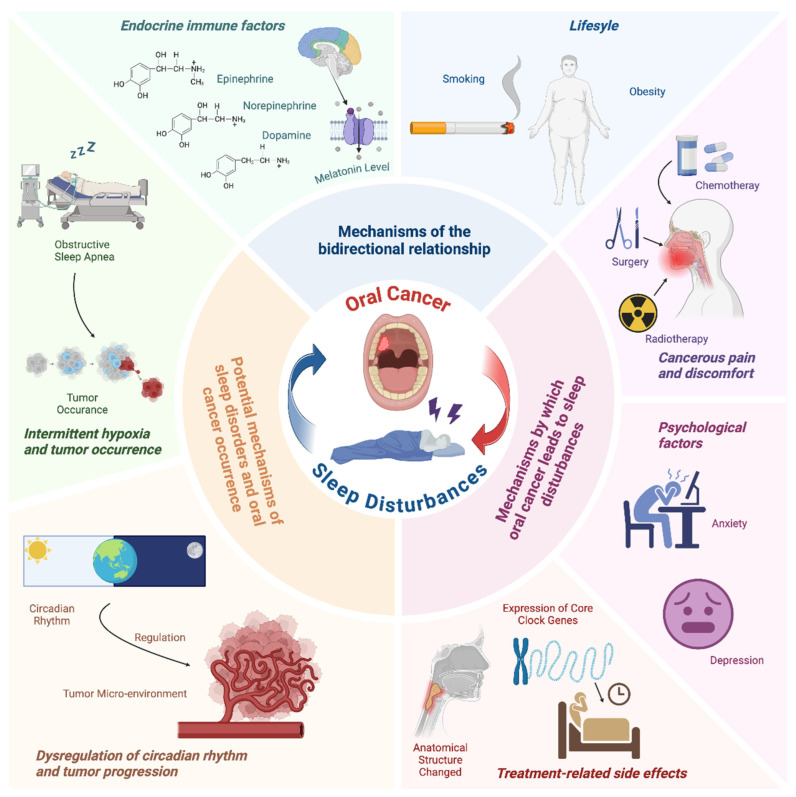
Mechanism of interaction between sleep disturbance and oral cancer.

## Data Availability

No new data were created or analyzed in this study.

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
