# Peer review of "Oral Cancer and Sleep Disturbances: A Narrative Review on Exploring the Bidirectional Relationship"

_cancers, 2025, doi:10.3390/cancers17081262_

Round 1
Reviewer 1 Report
Comments and Suggestions for Authors
I have no comments.
Authors should state this is a narrative review.
Author Response
Comments 1:Authors should state this is a narrative review.
Response 1:Thanks for your guidance! We agree with this comment. In response to the comment regarding the review type declaration, we have explicitly stated the nature of this work as a narrative review. The following statement has been added to the Title and Introduction section:
The title has been changed to "Oral Cancer and Sleep Disturbances: A Narrative Review on Exploring the Bidirectional Relationship."
In the introduction section, we have added:
This study is a narrative review that primarily integrates research evidence from the PubMed database on oral cancer and sleep disturbances from 1994 to 2025. The search strategy employed the following keyword combinations: "sleep disturbances," "obstructive sleep apnea syndrome," "oral cancer," "oral squamous cell carcinoma," "etiology of oral cancer," "etiology of sleep disturbances," "treatment of oral cancer," and "treatment of sleep disturbances." Literature selection primarily focuses on studies published in the last decade, with the research topic required to involve oral cancer (especially oral squamous cell carcinoma, OSCC) or sleep disturbances (such as obstructive sleep apnea, OSA), including their etiology, pathogenesis, treatment, and correlations. Additionally, early seminal studies of significant importance are selectively included. Studies that are unrelated to the research topic or of low methodological quality such as those with small sample sizes, flawed study designs, or unsupported conclusions are excluded.
Thank you again for this valuable suggestion to improve the manuscript's clarity.

Reviewer 2 Report
Comments and Suggestions for Authors
This review explores the bidirectional relationship between oral squamous cell carcinoma (OSCC) and sleep disturbances. Both diseases are crucial for patients, and their interrelation should be stressed.
The authors focused on mechanisms by which oral cancer leads to sleep disturbances, and vice versa, potential mechanisms underlying the correlation between sleep disorders and OSCC occurrence. In further units, other factors that prove a bidirectional relationship were discussed and possible treatment and prevention were proposed.
In my opinion, the problems discussed in this manuscript are interesting and not commonly known. Thus, the paper is worth attention. The required Figures have been prepared.
However, some minor issues must be improved:
- Please include additional 3-4 keywords, e.g., obstructive sleep apnea and head and neck cancer.
- Gene names should be written in italics.
- In line 243, the abbreviation HNSCC was used - please add a full name.
- The order of Figure numbers is incorrect: Figure 2 and next Figure 1. Please correct, remembering the numbers mentioned in the main text.
- Please prepare a References list according to the Jurnal style of citation.
Author Response
Comments 1:Please include additional 3-4 keywords, e.g., obstructive sleep apnea and head and neck cancer.
Response 1:Thanks for your guidance! We agree with this comment. Regarding the addition of keywords, the term "head and neck cancer" has been uniformly corrected to "oral cancer" throughout the manuscript. Therefore, the revised keywords are: Oral Cancer; Sleep Disturbances; Bidirectional Relationship; Obstructive Sleep Apnea; Intermittent Hypoxia; Disruption Of Circadian Rhythms;
Comments 2: Gene names should be written in italics.
Response 2:Thanks for your guidance! We agree with this comment.All gene names have been corrected to italics.
Comments 3:In line 243, the abbreviation HNSCC was used - please add a full name.
Response 3:Thank you for your guidance! We agree with this comment. In 3.2. Dysregulation of circadian rhythm and tumor progression, HNSCC has been corrected to Oral cancer.
Comments 4: The order of Figure numbers is incorrect: Figure 2 and next Figure 1. Please correct, remembering the numbers mentioned in the main text.
Response 4:Thank you for your guidance! We agree with this comment. The order of the figures has been corrected.
Comments 5: Please prepare a References list according to the Jurnal style of citation.
Response 5:Thank you for your guidance! We agree with this comment. The references have been updated.

Reviewer 3 Report
Comments and Suggestions for Authors
In this review, the authors explored the intricate, bidirectional relationship between oral cancer and sleep disturbances, particularly obstructive sleep apnea (OSA). They examined how factors such as pain, psychological stress, and treatment-related side effects contribute to sleep disorders in oral cancer patients. Additionally, they analyzed how sleep disturbances, through mechanisms like chronic inflammation, intermittent hypoxia, oxidative stress, and circadian rhythm disruption, may accelerate oral cancer progression. By integrating these insights, the authors provided a theoretical foundation for a more holistic approach to treatment, emphasizing the need for interdisciplinary strategies that address both tumor management and sleep optimization to improve patient outcomes. I have few suggestions
- The introduction and abstract rely on several key studies, but it would strengthen the argument to include additional supporting evidence regarding mechanisms linking sleep disorders to oral cancer. For instance, the role of chronic inflammation and oxidative stress in tumor progression could be discussed with specific molecular pathways.
- The review mentions analyzing PubMed studies from 1994 to 2025 but does not specify the selection criteria, inclusion/exclusion parameters, or search terms. A brief methodology section detailing these aspects would improve transparency and reproducibility.
- The manuscript could benefit from a stronger emphasis on potential clinical applications. Are there specific interventions, screening methods, or treatment modifications that could help mitigate the sleep-related complications in oral cancer patients?
- Minor grammatical issues such as “This review provided a theoretical foundation…” (should be "provides" instead of "provided" for consistency in tense) should be corrected.
- Consider restructuring certain sections for better flow. For example, in 1 Tumor size and location, the transition from describing tumor locations to discussing OSA could be smoother. You might start with the link between oral cancer and OSA before detailing anatomical changes.
- In 2 Cancerous pain and discomfort, the discussion of TNF-α, PAR2, and BDNF is insightful, but a brief explanation of how these molecular pathways contribute to sleep disturbances would enhance readability for a broader audience.
- Ensure uniformity in referring to sleep disturbances, OSA, and other related conditions. Some sections focus on OSA specifically, while others discuss general sleep disturbances without explicitly linking them to OSA, which might cause confusion.
- The discussion of NF-κB and cytokine signaling (lines 217–219) could be expanded to include how these pathways specifically contribute to oral cancer progression, such as their role in epithelial-to-mesenchymal transition (EMT) or immune evasion.
- The impact of circadian disruption on CSCs (lines 247–249) is interesting but could be linked to specific pathways or molecular markers for clarity.
- The discussion of melatonin (lines 270–285) is detailed but could benefit from a clearer transition linking it back to sleep disorders. Additionally, it would be useful to mention whether melatonin supplementation has been studied as a potential therapeutic strategy for oral cancer patients.
- The text explains smoking-induced carcinogenesis well but could elaborate further on the specific pathways linking obesity to oral cancer. While IGF and chronic inflammation are mentioned, expanding on additional mechanisms (e.g., adipokine-mediated immune modulation) would strengthen the discussion.
- The section on pain management is well-structured but could benefit from more detail on the comparative effectiveness of different treatments. For example, while CPAP is mentioned as the preferred OSA treatment, a brief discussion on patient adherence challenges and alternative interventions (e.g., lifestyle modifications) would be valuable.
Author Response
Comments 1: The introduction and abstract rely on several key studies, but it would strengthen the argument to include additional supporting evidence regarding mechanisms linking sleep disorders to oral cancer. For instance, the role of chronic inflammation and oxidative stress in tumor progression could be discussed with specific molecular pathways.
Response 1:Thanks for your guidance! We agree with this comment. In response to the supplement on the mechanisms linking sleep disturbances and oral cancer, we have added the following content to the third paragraph of the introduction:
The primary pathophysiological mechanisms of OSA include chronic inflammation and oxidative stress, both of which are critical factors influencing the development and progression of oral cancer. Chronic inflammation is strongly associated with the initiation, metastasis, and invasion of oral cancer[11], likely through interactions among various pro-inflammatory mediators. For instance, epidermal growth factor (EGF) can induce epithelial-mesenchymal transition (EMT) in oral cancer cells via the Akt/Ezrin Tyr353/NF-κB pathway, thereby enhancing their motility and invasiveness[12]. Multiple studies have identified matrix metalloproteinases (MMPs) as key mediators of oral cancer progression. Specifically, MMP-11 has been linked to lymph node metastasis and increased malignancy, with high MMP-11 expression correlating with significantly reduced survival rates in oral cancer patients[13]. The inflammatory cytokine interleukin-11 (IL-11)/gp130 can upregulate MMP-13 expression in oral squamous cell carcinoma (OSCC) through activation of the PI3K/Akt and AP-1 signaling pathways, thereby promoting cancer cell migration[14]. Furthermore, oxidative stress contributes to oral cancer development by promoting M1-like tumor-associated macrophage polarization via Thbs1-mediated mechanisms[15].
Comments 2: The review mentions analyzing PubMed studies from 1994 to 2025 but does not specify the selection criteria, inclusion/exclusion parameters, or search terms. A brief methodology section detailing these aspects would improve transparency and reproducibility.
Response 2:Thanks for your guidance! We agree with this comment. In response to the selection criteria, The following statement has been added to the Introduction section:
This study is a narrative review that primarily integrates research evidence from the PubMed database on oral cancer and sleep disturbances from 1994 to 2025. The search strategy employed the following keyword combinations: "sleep disturbances," "obstructive sleep apnea syndrome," "oral cancer," "oral squamous cell carcinoma," "etiology of oral cancer," "etiology of sleep disturbances," "treatment of oral cancer," and "treatment of sleep disturbances." Literature selection primarily focuses on studies published in the last decade, with the research topic required to involve oral cancer (especially oral squamous cell carcinoma, OSCC) or sleep disturbances (such as obstructive sleep apnea, OSA), including their etiology, pathogenesis, treatment, and correlations. Additionally, early seminal studies of significant importance are selectively included. Studies that are unrelated to the research topic or of low methodological quality such as those with small sample sizes, flawed study designs, or unsupported conclusions are excluded.
Comments 3: The manuscript could benefit from a stronger emphasis on potential clinical applications. Are there specific interventions, screening methods, or treatment modifications that could help mitigate the sleep-related complications in oral cancer patients.
Response 3:Thanks for your guidance! We agree with this comment. In response to supplement the potential clinical applications, we have added the following content in 5.1. Lifestyle Management and Psychobehavioral Therapy section of the article:
Studies have shown that a 12-week exercise intervention, including strength training, resistance training, walking, cycling, yoga, qigong, or tai chi, can significantly improve sleep disorders, anxiety, pain, and fatigue in patients with oral cancer[81]. Another study also indicated that regular physical activity interventions help improve cardiopulmonary health, fatigue levels, and overall quality of life in oral cancer survivors[82]. Enhancing physical activity may be a key approach to improving sleep quality in oral cancer patients.
Comments 4: Minor grammatical issues such as “This review provided a theoretical foundation…” (should be "provides" instead of "provided" for consistency in tense) should be corrected.
Response 4:Thanks for your guidance! We agree with this comment. In response to the grammatical issues, we have made corrections accordingly. Thank you again for your suggestions.
Comments 5: Consider restructuring certain sections for better flow. For example, in 1 Tumor size and location, the transition from describing tumor locations to discussing OSA could be smoother. You might start with the link between oral cancer and OSA before detailing anatomical changes.
Response 5:Thanks for your guidance! We agree with this comment.In response to the structural adjustments, we have supplemented the relationship between OSA and oral cancer in 2.1 Tumor size and location section:
Sleep disorders are commonly observed in patients with oral cancer. Studies have shown that among 412 patients diagnosed with oral cancer, approximately one-third experience persistently poor or worsening sleep quality. Moreover, the prevalence of obstructive sleep apnea (OSA) in this population is as high as 91.7%, significantly higher than in the general population. This phenomenon may be associated with the tumor's size and anatomical location.
Comments 6: In 2 Cancerous pain and discomfort, the discussion of TNF-α, PAR2, and BDNF is insightful, but a brief explanation of how these molecular pathways contribute to sleep disturbances would enhance readability for a broader audience.
Response 6: Thanks for your guidance! We agree with this comment. In response to the issue of insufficient elaboration on the relationship between pain and sleep disorders, we have supplemented this content in the "2.2. Cancerous pain and discomfort " section:
Studies have shown that severe and persistent chronic pain often leads to a decline in sleep quality[26]. Moreover, patients suffering from chronic facial pain tend to experience reduced sleep duration, lower health-related quality of life, and increased levels of stress and fatigue[27].
Comments 7: Ensure uniformity in referring to sleep disturbances, OSA, and other related conditions. Some sections focus on OSA specifically, while others discuss general sleep disturbances without explicitly linking them to OSA, which might cause confusion.
Response 7: Thanks for your guidance! We agree with this comment. Corrections have been made throughout the text. After the term "sleep disturbance" or "sleep disorder", the specific type is indicated, such as "sleep disturbance, such as insomnia" or "sleep disorder, such as insomnia" or "sleep disorder, especially OSA," to distinguish general sleep disorders from OSA and avoid confusion.
Comments 8: The discussion of NF-κB and cytokine signaling (lines 217–219) could be expanded to include how these pathways specifically contribute to oral cancer progression, such as their role in epithelial-to-mesenchymal transition (EMT) or immune evasion.
Response 8:Thank you for your guidance! We agree with this comment. Regarding the role of NF-κB in EMT, In 3.1. Intermittent hypoxia and tumor occurrence, we have supplemented the content as follows:
Additionally, NF-κB is a critical regulator of metastasis in oral cancer, primarily mediating the epithelial-mesenchymal transition (EMT) in cancer cells. Studies have shown that ZEB1/2, Transforming Growth Factor-beta (TGF-β), and Slug act as EMT inducers upregulated by NF-κB. Following NF-κB-driven EMT induction, E-cadherin levels are significantly reduced, while N-cadherin and vimentin levels are markedly elevated, thereby promoting the metastasis of tumor cells.
Comments 9: The impact of circadian disruption on CSCs (lines 247–249) is interesting but could be linked to specific pathways or molecular markers for clarity.
Response 9:Thank you for your guidance! We agree with this comment. To supplement the specific mechanisms of the circadian rhythm in oral cancer stem cells, In 3.2. Dysregulation of circadian rhythm and tumor progression, we provide the following additional content:
The specific mechanism is as follows: the core clock gene Period 1 (PER1) can inhibit glycolysis-mediated cell proliferation by forming the PER1/RACK1/PI3K complex, thereby suppressing the progression of oral cancer cells. Another study also suggests that the clock gene Per2 plays a crucial role in the balance of cell cycle progression, proliferation, and apoptosis by regulating the cyclin/CDK/CKI cell cycle network.
Comments 10:The discussion of melatonin (lines 270–285) is detailed but could benefit from a clearer transition linking it back to sleep disorders. Additionally, it would be useful to mention whether melatonin supplementation has been studied as a potential therapeutic strategy for oral cancer patients.
Response 10:Thank you for your guidance! We agree with this comment. Regarding the role of melatonin in sleep disorders and its therapeutic effects on oral cancer patients, we have supplemented Section 4.1. Endocrine immune factors with the following content:
The effects of melatonin in inhibiting cancer cell proliferation, promoting autophagy, and inducing apoptosis not only contribute to the suppression of oral cancer progression but may also influence the overall physiological state of the body, particularly sleep regulation. As a key hormone regulating the sleep-wake cycle, abnormal melatonin levels are often closely associated with sleep disorders. Studies have shown that exogenous melatonin supplementation can be used to treat primary and secondary sleep disorders, reduce sleep latency in primary insomnia, and improve delayed sleep phase syndrome, thereby optimizing the sleep-wake pattern. Given that oral cancer patients often experience reduced sleep quality, circadian rhythm disturbances, and impaired immune function, appropriate melatonin supplementation may not only improve their sleep conditions but also exert anti-inflammatory, immunomodulatory, and antitumor effects. Cellular experiments have demonstrated that melatonin, in combination with the ferroptosis inducer erastin, can exert a synergistic antitumor effect on oral squamous cell carcinoma by inducing apoptosis, ferroptosis, and inhibiting autophagy through ROS promotion. Additionally, research has found that exogenous supplementation of melatonin and vitamin D can help address antioxidant imbalance in oral cancer. Melatonin supplementation may inhibit tumor growth, improve survival rates and quality of life, and enhance the efficacy of radiotherapy, thereby providing an adjunctive strategy for oral cancer treatment.
Comments 11: The text explains smoking-induced carcinogenesis well but could elaborate further on the specific pathways linking obesity to oral cancer. While IGF and chronic inflammation are mentioned, expanding on additional mechanisms (e.g., adipokine-mediated immune modulation) would strengthen the discussion.
Response 11:Thank you for your guidance! We agree with this comment. Regarding the immunoregulation mediated by adipokines, we have supplemented section 4.2 Lifestyle as follows:
The adipokine Chemerin can enhance the invasiveness of oral cancer cells by activating the STAT3 signaling pathway and promoting the production of IL-6 and TNF-α. Additionally, high-fat diet (HFD) induced obesity significantly promotes the occurrence of oral cancer. The underlying mechanism may involve obesity recruiting CD11bGr1 myeloid-derived suppressor cells (MDSCs) through the CCL9/CCR1 axis, thereby altering the local immune microenvironment and enhancing the immunosuppressive function of MDSCs via intracellular fatty acid uptake.
Comments 12: The section on pain management is well-structured but could benefit from more detail on the comparative effectiveness of different treatments. For example, while CPAP is mentioned as the preferred OSA treatment, a brief discussion on patient adherence challenges and alternative interventions (e.g., lifestyle modifications) would be valuable
Response 12:Thank you for your guidance! We agree with this comment. Regarding the discussion on CPAP adherence and alternative interventions, we have supplemented section 5.2 Pain management and OSA treatment on pain management as follows:
Due to subjective discomfort, including dry mouth and chest discomfort, CPAP adherence is poor. Even among patients who initiate CPAP therapy, 50% discontinue its use within the first year. Therefore, finding alternative treatments for CPAP is crucial. Currently, orofacial myofunctional therapy (OMT), which includes isotonic and isometric exercises targeting the oral and oropharyngeal structures, has been clinically adopted. OMT can improve clinical symptoms in OSA patients by enhancing muscle tone, endurance, and coordination of the pharyngeal and peripharyngeal muscles. Benefits include reducing snoring, lowering the apnea-hypopnea index, and improving daytime sleepiness and sleep quality. Additionally, positional therapy, which involves devices that prevent the supine position during sleep by inducing neck or chest vibrations, has shown significant efficacy in OSA patients who experience more frequent and severe respiratory events while sleeping in the supine position (Postural related sleep apnea).

Reviewer 4 Report
Comments and Suggestions for Authors
TITLE: Oral Cancer and Sleep Disturbances: Exploring the Bidirectional Relationship
Title:
Could you include in your title which kind of study is your article?
Abstract:
Nothing to state.
Introduction:
You already performed the review, so you must include your verbs in the past tense.
You used two terms: oral cancer and head and neck cancer. This must be clarified. Your study is interested in which type of cancer? Or both?
Methods:
Where are your methods? How did you perform your review? This is essential. Even if it’s not a systematic review, you must include the methods of your review.
Results:
I understand that, since point 2, these are the results of your search, so I proceed.
I think you performed an interesting summary of proved and potential mechanisms of relationship.
Which is the location where patients usually present with pain? Does this interfere with, for example, nutrition?
Please, you must include figure 1 first, then figure 2. Change the order.
Discussion:
Where are the limitations of your study? Are there biases in your study? Are there biases in the literature? This must be included in your study.
You discussed the correlation between OSA and oral cancer. However, is this an association, or there is a causal relationship? Could you discuss this? Are there longitudinal studies discussing this fact?
Could you include some proposal for future directions of research? You included, together with conclusions, some interesting point, but it seems a bit general. Which facts could be interesting to be studied?
Are there studies proving the effectiveness of sleep disorders in oral cancer? Or viceversa?
What about nutrition? Is this important in these patients? You mentioned some factors, such as smoking, but you should include more possible factors and present a more holistic point of view.
Author Response
Comments 1: Title:Could you include in your title which kind of study is your article?
Response 1:Thanks for your guidance! We agree with this comment. In response to the comment regarding the review type declaration, we have explicitly stated the nature of this work as a narrative review. The following statement has been added to the Title section:
The title has been changed to "Oral Cancer and Sleep Disturbances: A Narrative Review on Exploring the Bidirectional Relationship."
Comments 2: Where are your methods? How did you perform your review? This is essential. Even if it’s not a systematic review, you must include the methods of your review.
Response 2:Thanks for your guidance! We agree with this comment. In response to the methods. The following statement has been added to the Introduction section:
This study is a narrative review that primarily integrates research evidence from the PubMed database on oral cancer and sleep disturbances from 1994 to 2025. The search strategy employed the following keyword combinations: "sleep disturbances," "obstructive sleep apnea syndrome," "oral cancer," "oral squamous cell carcinoma," "etiology of oral cancer," "etiology of sleep disturbances," "treatment of oral cancer," and "treatment of sleep disturbances." Literature selection primarily focuses on studies published in the last decade, with the research topic required to involve oral cancer (especially oral squamous cell carcinoma, OSCC) or sleep disturbances (such as obstructive sleep apnea, OSA), including their etiology, pathogenesis, treatment, and correlations. Additionally, early seminal studies of significant importance are selectively included. Studies that are unrelated to the research topic or of low methodological quality such as those with small sample sizes, flawed study designs, or unsupported conclusions are excluded.
Comments 3: Results:I understand that, since point 2, these are the results of your search, so I proceed.I think you performed an interesting summary of proved and potential mechanisms of relationship. Which is the location where patients usually present with pain? Does this interfere with, for example, nutrition?
Response 3:Thanks for your guidance! We agree with this comment. Regarding the questions about the location and impact of pain, we have added the following content in the "2.2 Cancerous Pain and Discomfort" section:
For oral cancer patients, orofacial pain is one of the earliest and most severe symptoms, impairing their speech, swallowing, eating, and drinking functions.
Comments 4: Please, you must include figure 1 first, then figure 2. Change the order.
Response 4:Thank you for your guidance! We agree with this comment. The order of the figures has been corrected.
Comments 5: Discussion:Where are the limitations of your study? Are there biases in your study? Are there biases in the literature? This must be included in your study.
Response 5:Thank you for your guidance! We agree with this comment. In response to the discussion on the limitations of the article, the following content has been added to the discussion section:
However, this narrative review has certain limitations. First, the study relies on previously published literature, which may introduce selection and publication bias. The heterogeneous nature of the included studies, including differences in study design, sample population, and sleep assessment tools, poses a challenge for direct comparisons. Additionally, most studies are retrospective or observational, limiting causal inference. Biases in the included studies, such as recall bias, confounding factors, and small sample sizes, may also affect the generalizability of the findings. Future research should focus on large-scale prospective cohort studies with objective sleep assessment and better control for confounders to establish a clearer causal relationship between oral cancer and sleep disturbances.
Comments 6: You discussed the correlation between OSA and oral cancer. However, is this an association, or there is a causal relationship? Could you discuss this? Are there longitudinal studies discussing this fact?
Response 6:Thank you for your guidance! We agree with this comment. As a supplement to the discussion on the relationship between OSA and sleep disorders, we have added the following content in the discussion section:
Current evidence suggests a potential association between obstructive sleep apnea (OSA) and oral cancer; however, a direct causal relationship remains unclear. This association may be mediated through various mechanisms, including chronic intermittent hypoxia, oxidative stress, and inflammatory responses. However, the presence of shared risk factors, such as smoking, obesity, and chronic inflammation, complicates the determination of causality.
Regarding longitudinal studies, while some prospective cohort studies have indicated an increased incidence of certain cancers among OSA patients, specific data on oral cancer remain limited. Long-term follow-up studies are needed to establish a causal relationship. Future research should focus on large-scale longitudinal cohorts and interventional trials to explore whether OSA treatments, such as continuous positive airway pressure (CPAP), can reduce cancer risk.
Comments 7: Could you include some proposal for future directions of research? You included, together with conclusions, some interesting point, but it seems a bit general. Which facts could be interesting to be studied?
Response 7:Thank you for your guidance! We agree with this comment. Regarding the addition to future research directions, we have included the following content in the discussion section:
Future research should focus on the following key areas: At the mechanistic level, further investigations are needed to elucidate the regulatory pathways through which circadian rhythm disruption, inflammatory responses, immune dysregulation, and hormonal fluctuations influence tumor growth and metastasis. Large-scale, multicenter prospective cohort studies should be designed to establish the causal and temporal relationships between sleep disturbances and the incidence or prognosis of oral cancer. Additionally, interventional clinical trials are essential to assess the impact of sleep optimization strategies—such as continuous positive airway pressure (CPAP) therapy and cognitive behavioral therapy—on tumor progression. Epigenetic mechanisms warrant particular attention, with a focus on exploring sleep disturbance-related alterations in DNA methylation and histone modifications as potential biomarkers. From a translational perspective, developing risk prediction models that integrate genetic profiles, metabolic phenotypes, and lifestyle factors will be crucial for advancing personalized sleep management. Moreover, a comprehensive evaluation of sleep interventions on patients' psychological well-being and quality of life is necessary to establish a multidimensional bio-psycho-social support system.
Comments 8:Are there studies proving the effectiveness of sleep disorders in oral cancer? Or viceversa?
Response 8:Thank you for your guidance! We agree with this comment. Currently, studies specifically investigating the role of sleep disturbances in oral cancer are limited. However, existing evidence suggests that sleep disturbances may influence cancer development and progression through mechanisms such as inflammation, immune regulation, hormonal imbalances (e.g., cortisol and melatonin alterations), and oxidative stress. Conversely, oral cancer and its treatments (e.g., surgery, radiotherapy, and chemotherapy) can contribute to sleep disturbances due to factors like pain, anxiety, depression, and structural changes in the upper airway. Research indicates that cancer patients often experience poor sleep quality, which may impact disease prognosis and quality of life.
Comments 9:What about nutrition? Is this important in these patients? You mentioned some factors, such as smoking, but you should include more possible factors and present a more holistic point of view.
Response 9:Thank you for your guidance! We agree with this comment. Regarding the supplementation on the role of nutrition, we have added the following content in section 4.2 "Lifestyle." Furthermore, concerning the influencing factors, in addition to smoking, other important factors such as nutrition and obesity also affect the progression of oral cancer and are associated with sleep disturbances. This has been addressed in the manuscript as follows:
Nutrition plays a crucial role in both oral cancer and obstructive sleep apnea (OSA). Patients with oral cancer often face the risk of inadequate nutrient intake due to the presence of the tumor itself and the side effects that may arise during treatment, such as oral ulcers, dry mouth, taste alterations, nausea, and loss of appetite caused by surgery, radiotherapy, or chemotherapy. Additionally, difficulties in chewing and swallowing further limit their ability to consume essential nutrients, making them more susceptible to malnutrition. Studies have shown that nutritional and physical prehabilitation interventions can improve the overall health status of oral cancer patients and positively impact clinical outcomes. Malnutrition not only leads to weight loss, impaired immune function, and muscle wasting but may also negatively affect sleep quality. For instance, vitamin D deficiency has been associated with an increased risk of OSA. The underlying mechanism may involve vitamin D receptors and the enzymes that regulate its activation and degradation, which are implicated in sleep regulation in the brain. Furthermore, vitamin D plays a role in the pathway for melatonin production, a hormone that regulates the human circadian rhythm and sleep. In addition, vitamin D may indirectly influence sleep through its effects on nonspecific pain disorders and has been linked to obstructive sleep apnea syndrome.

Round 2
Reviewer 4 Report
Comments and Suggestions for Authors
Thank you for your changes and modifications